# Melanin and Neuromelanin Fluorescence Studies Focusing on Parkinson’s Disease and Its Inherent Risk for Melanoma

**DOI:** 10.3390/cells8060592

**Published:** 2019-06-15

**Authors:** Dieter Leupold, Lukasz Szyc, Goran Stankovic, Sabrina Strobel, Hans-Ullrich Völker, Ulrike Fleck, Thomas Müller, Matthias Scholz, Peter Riederer, Camelia-Maria Monoranu

**Affiliations:** 1LTB Lasertechnik Berlin GmbH, 12489 Berlin, Germany; dieter.leupold@ltb-berlin.de (D.L.); matthias.scholz@ltb-holding.de (M.S.); 2Magnosco GmbH, 12489 Berlin, Germany; lukasz.szyc@magnosco.com (L.S.); goran.stankovic@magnosco.com (G.S.); 3Institute of Pathology, Department of Neuropathology, University of Wuerzburg, Comprehensive Cancer Center (CCC) Mainfranken Wuerzburg, 97080 Wuerzburg, Germany; sabrina.strobel@uni-wuerzburg.de; 4Pathology, Leopoldina Krankenhaus GmbH, Gustav-Adolf-Str 8, D-97422 Schweinfurt, Germany; hvoelker@leopoldina.de; 5Dermatology Practice, 10117 Berlin, Germany; kontakt@hautarzt-fleck.de; 6Department of Neurology, St. Joseph Hospital Berlin-Weißensee, 13088 Berlin, Germany; Th.Mueller@alexianer.de; 7Center of Mental Health, Department of Psychiatry, Psychosomatics and Psychotherapy, Margarete-Hoeppel-Platz 1, 97080 Wuerzburg, Germany; peter.riederer@mail.uni-wuerzburg.de; 8Department and Research Unit of Psychiatry, University of Southern Denmark, Odense, Odense C - DK-5000, Denmark

**Keywords:** Parkinson’s disease, melanin, neuromelanin, dermatofluoroscopy

## Abstract

Parkinson’s disease is associated with an increased risk of melanoma (and vice versa). Several hypotheses underline this link, such as pathways affecting both melanin and neuromelanin. For the first time, the fluorescence of melanin and neuromelanin is selectively accessible using a new method of nonlinear spectroscopy, based on a stepwise two-photon excitation. Cutaneous pigmentation and postmortem neuromelanin of Parkinson patients were characterized by fluorescence spectra and compared with controls. Spectral differences could not be documented, implying that there is neither a Parkinson fingerprint in cutaneous melanin spectra nor a melanin-associated fingerprint indicating an increased melanoma risk. Our measurements suggest that Parkinson’s disease occurs without a configuration change of neuromelanin. However, Parkinson patients displayed the same dermatofluorescence spectroscopic fingerprint of a local malignant transformation as controls. This is the first comparative retrospective fluorescence analysis of cutaneous melanin and postmortem neuromelanin based on nonlinear spectroscopy in patients with Parkinson’s disease and controls, and this method is a very suitable diagnostic tool for melanoma screening and early detection in Parkinson patients. Our results suggest a non-pigmentary pathway as the main link between Parkinson’s disease and melanoma, and they do not rule out the melanocortin-1-receptor gene as an additional bridge between both diseases.

## 1. Introduction

The available epidemiological data regarding a possible association between cutaneous malignant melanoma (MM) and Parkinson’s disease (PD) show a two-way relationship. On the one hand, patients with PD have significantly higher risk for MM [1,2,3], while, on the other hand, patients with MM carry an increased risk of PD [1,3]. In order to understand this link between PD and MM, three major hypotheses were postulated over the past decades, as outlined below.

Firstly, regarding the potential role of pigmentation as a possible bridge between PD and MM, a first aspect to be considered is the possibility of disorders within the main gene which regulates pigmentation, melanocortin 1 receptor (*MC1R*). *MC1R* is responsible for constitutive variation of pigmentation in humans. Specific *MC1R* variants may predispose to both MM and PD [4,5]. There are indications in relevant literature that loss-of-function *MC1R* variants, which result in disturbed melanogenesis, are associated with higher risk of developing MM [6]. Several other authors discussed an association between PD and MM due to *MC1R* variants [3,7], and most recent results relating to the role of *MC1R* in dopaminergic neurons led to the conclusion that *MC1R* may represent a common pathogenic pathway for MM and PD [8].

A second focus is directed toward α-synuclein. A pathological structural change in α-synuclein is a hallmark of PD [9]. This protein is present in catecholamine-containing neurons of the substantia nigra (SN) and locus coeruleus (LC) and binds to neuromelanin (NM). Current studies indicate that α-synuclein may affect the biosynthesis of melanin and NM by regulating activities of certain enzymes, e.g., tyrosinase, tyrosine hydroxylase, and peroxidase [6]. In addition, pathological α-synuclein may change the biosynthesis of NM. There are also indications of structural changes in NM associated with α-synuclein aggregation [10].

In normal epidermal melanocytes, no α-synuclein is found [11], but it is expressed in cutaneous melanocytic lesions, as can be demonstrated for melanoma cell lines and nevus tissue [11]. Here, also due to its possible interaction with tyrosinase [12], α-synuclein may even play a role in regulating the biosynthesis of melanin in skin. There are research approaches concerning the use of α-synuclein as a biomarker for melanomagenesis [13]. Therefore, in PD patients, the biosynthesis of cutaneous melanin may be modified. More recently, α-synuclein was detected in skin nerves from PD patients [14,15]. While α-synuclein cannot be regarded as a selective skin biomarker for the early detection of PD, a combination of skin nerve p-α-synuclein detection with an analysis of rapid eye movement (REM) sleep behavior was proposed to detect PD in its prodromal state [15].

Lastly, recent studies focused on the tumor suppressor gene *PARK2*, which synthesizes the protein Parkin [16,17]. Their results indicate that Parkin inactivation (loss-of-function mutations in *PARK2*), which is often found in young-onset PD, may also strengthen a predisposition to and progression of melanoma [18].

The first part of the following investigation is directed toward the cutaneous pigmentation of PD patients with special emphasis on a potentially disturbed melanogenesis, and the underlying question of whether a fingerprint of PD and/or a fingerprint of an increased MM risk can be identified in cutaneous melanin. All discussed aspects of the PD–MM link may affect the biosynthesis of cutaneous melanin. Such disturbed melanogenesis may concern both structural changes in the melanin subunits (e.g., 5,6–dihydroxyindole (DHI) and 5,6–dihydroxyindole-2-carboxylic acid (DHICA) in eumelanin) or their arrangement in the melanin polymer, as well as changes in the melanin microenvironment in melanosomes (see below). The present investigation aims to examine such possible changes in pigmentation of PD patients’ skin directly by dermatofluoroscopy. This examination involves both normal pigmented tissue and specimens from nevi. A special focus was placed on signs of incipient malignant melanocytic degeneration, for which dermatofluoroscopy is particularly suitable [19,20]. None of the PD patients studied were current melanoma patients, but two of them were in dermatological surveillance. Thus, in addition to benign nevi, clinically atypical (dysplastic) nevi could be included. Since this investigation is directed toward a possible pigmentary pathway with changed biosynthesis of melanin and/or NM being part of the linkage between PD and MM, we included the NM of the substantia nigra pars compacta (SNpc) of the human brain into our investigation, in addition to the pigmentation of the skin as the site of cutaneous MM. To our knowledge, no fluorescence spectra excited by ultraviolet (UV) or visible radiation were previously found for NM in situ. However, a weak fluorescence of synthetic NM was reported, enhanced by nicotine binding [21]. Since NM also consists of eu- and pheomelanin [22], the same method of stepwise two-photon excitation was used to obtain fluorescence spectra.

## 2. Material and Methods

### 2.1. Introductory Overview

The present work is based on a retrospective investigation carried out on PD patients as part of routine survival as a melanoma check-up. This check-up is indicated because of the increased melanoma risk of Parkinson’s patients. These investigations were carried out with informed written consent of the patients and were advertised according to § 4 paragraph 23 clause 3 (German Medicine Law) at the medical association. It was characterized as non-interventional and, thus, observational because all performed evaluations are part of routine surveillance in the treatment of PD patients.

For these investigations, the diagnostic device for melanoma, derma FC (magnosco GmbH, Berlin, Germany), was used. The derma FC is a European Commission (EC)-certified medical device, approved for the European Market Class IIa (EC Certificate Directive 93742/EEC, Annex II excluding. Full Quality Assurance System, Certificate Registration No.: Z-16-213-SZ-RII). This certificate is based on a prospective multicenter clinical trial involving 620 lesions suspicious for melanoma. The device is approved for patients of the Caucasian skin type. As a result of the measurement, the device indicates in the form of a numerical value (score) whether the investigation speaks for a melanoma or a benign or dysplastic nevus. It is a decision aid for the final diagnosis of the clinician. This score is determined from the melanin fluorescence spectra from hundreds of microareas distributed throughout the pigmented lesion. The grid pitch is 200 µm. The diameter of each analyzed microarea is 30 µm. The underlying evaluation method is based on dermatofluoroscopy (see below). These fluorescence spectra are stored in the device together with the displayed examination result. They are not usually of concern to the clinician, but may be informative for in-depth questions (such as follow-up). The results presented here are based on the retrospective analysis of such melanin fluorescence spectra from 19 pigmented lesions of 14 PD patients.

### 2.2. Melanin Fluorescence

Dermatofluoroscopy was developed with the aim of becoming a new diagnostic tool for early detection of melanotic malignant melanoma [19,20,23]. It is based on the measurement of the spectral distribution of ultra-weak melanin fluorescence from the melanosomes of cutaneous pigmented cells. With conventional one-photon excitation, this melanin fluorescence is completely concealed by the much more intense fluorescence of other endogenous fluorophores, e.g., of NAD(P)H and flavins (the so-called autofluorescence of skin). To overcome this problem, a special fluorescence excitation mechanism of non-linear spectroscopy is used: a stepwise two-photon excitation of melanin with nanosecond pulses at 800 nm. This is the hallmark of dermatofluoroscopy. This type of excitation is unique for melanin in contrast to all other endogenous fluorophores because only melanin can absorb 800-nm photons. However, the energy of a single 800-nm photon is too low to excite fluorescence in the visible spectral range; this requires the subsequent absorption of a second 800-nm photon in the already excited state (for details, see below and Figure 1). The parameters of the excitation laser are chosen to prevent a simultaneous absorption of two 800-nm photons, as this would be more or less equivalent to a single 400-nm photon absorption and result in (unwanted) autofluorescence.

Figure 2 illustrates the energy level scheme of melanin (Jablonski diagram) and the underlying processes which determine the fluorescence spectrum. After absorption of the first photon, a first excited state is reached where there is competition between the (“downhill”) relaxation (k (N) in the nevus, k (MM) in the melanoma) and the strength of the second (“uphill”) absorption (determined by absorption cross-section and photon flux density of the excitation). Absorption cross-section and relaxation rate are quantities which sensitively depend on fluorophore electron structure and the microenvironment.

The spectrum of normally pigmented skin (Figure 3a) is a superposition of melanin fluorescence with a maximum at about 500 nm with the NAD(P)H-dominated autofluorescence (maximum at about 470 nm, obtained from oculo-cutaneous skin tissue). The ratio of the two components determines the position of the resulting fluorescence maximum [20]; on low pigmented Fitzpatrick skin type 1, both bands appear separately (D. Leupold et al., submitted). Melanin fluorescence from the nevomelanocytes of benign or dysplastic nevi (Figure 3b,c) shows a distinctly different, flatter, and red-shifted spectral profile than fluorescence from melanocytes. An obvious cause of the difference could be the lacking discharge of melanosoma out of melanocytes (shutdown of the dominant influence of keratinocytes) and the changing structural alignments of the melanin π-systems that determine fluorescence (“π-stacking”). The red shift of the fluorescence spectrum changes into the characteristic curve according to Figure 3d in melanoma: a constant increase in intensity from 440 nm to 650 nm. Mathematically described, this means that the first derivative of the spectral fluorescence course dI_F_ (lambda)/d (lambda) for melanoma is a horizontal straight line between 440 and 650 nm. The spectral fluorescence course of dysplastic nevi (Figure 3c) differs from that of melanoma by a reduced increase above about 570 nm (this means a drop in the constant value of the first derivative above 570 nm). Benign nevi are characterized by a zero crossing of the first derivative in the range between about 530 and 550 nm.

Such dermatofluoroscopic investigations were carried out on normal pigmented skin, and benign and dysplastic nevi, as well as melanomas of more than 500 patients of Caucasian origin. This resulted in a database of several tens of thousands of spectra, the vast majority of which can be assigned to one of the following four classes of melanin-dominated fluorescence spectra: melanosomes of melanocytes (Figure 3a, class 4), melanosomes of nevomelanocytes of benign nevi (Figure 3b, class 3), melanosomes of nevomelanocytes of dysplastic nevi (Figure 3c, class 2), and melanosomes of melanoma cells (Figure 3d, class 1). It is important to note that these four types of spectra capture all measured cutaneous melanin fluorescence. This implies that class 1 spectra display a fingerprint of melanoma cells, regardless of the melanoma subtype. This is valid for the main melanoma subtypes studied so far: in situ, superficial spreading, nodular, lentigo maligna, and acrolentiginous [24]. It also means that malignant melanocytic degeneration of a nevus toward melanoma is always represented by classes 2 and 1. The latter is also confirmed by our follow-up measurements of nevi over several years [25]. The automatic assignment of measured fluorescence spectra to one of these four classes is based on the minimum of the root-mean-squared difference (RMSD) between the measured curve and each of the model curves. The RMSD has a fixed upper limit, and spectra which exceed this limit elude this classification (e.g., hairs, marker fluorophores, or impurities). Further details of the automated assignment of the spectra measured with derma FC were previously described [23].

The spectral analysis is concentrated on the range between 430 nm and 650 nm. An increase of intensity below 430 nm results from second harmonic generation (SHG) in collagen. The signal in the range above 650 nm stems from a further nonlinear optical effect that it is not considered here.

### 2.3. Skin of PD Patients of Caucasian Origin

In total, 19 nevi, mainly located on the trunk and extremities, and adjacent normal pigmented skin areas of 14 PD patients (11 males, three females; mean age 62 years) from the Department of Neurology, Alexianer St. Joseph Hospital Berlin, Germany, were examined by dermatofluoroscopy. The patients were of Caucasian skin type/Fitzpatrick type 2–3 [26]. Except for red, all natural hair colors were represented. None of the patients had freckles. The skin areas to be examined were selected by a dermatologist. On full-body inspection, attention was placed on normal and dysplastic nevi. Among the investigated patient collective, no nevi were found that were clinically suspicious for being dysplastic; however, two of the selected patients are now in dermatological long-term observation. All PD patients were treated with a combination of monoamine oxidase B MAO-B-inhibitors and dopaminergic receptor agonists. An overview of the patients data is shown in Table 1.

All subjects gave written informed consent. This investigation was advertised according § 4 paragraph 23 clause 3 (German Medicine Law) at the medical association. It was characterized as non-interventional and, thus, observational because all performed evaluations are part of routine surveillance in the treatment of PD patients.

### 2.4. Postmortem Neuromelanin of Substantia Nigra

The investigations were directed toward the PD-relevant NM pigmentation of the SNpc of PD patients in relation to that of healthy controls. It is well known that, during progress of PD, there is a decrease in the concentration of neuromelanin, as well as indications of possible neuromelanin changes [27]. The present study was performed on histological specimens (formalin-fixed and paraffin-embedded, FFPE) of SNpc from deceased PD patients and controls. An overview of the patients data is shown in Table 2. Measurement of the fluorescence of the FFPE preparations of SNpc were also realized using the device derma FC (magnosco GmbH, Berlin, Germany). Similar to the dermal investigations, the SNpc area is covered with a measuring grid. The grid pitch is 100 microns, and the diameter of each spectroscopically analyzed brain area is 30 microns. In this way, several hundreds of spectra per investigated SNpc can be obtained. Formalin-fixed and paraffin-embedded postmortem brain tissue samples from the midbrain of 10 neuropathologically characterized PD cases with brainstem-predominant Braak stages 2–3, as well as 10 age-matched controls, were provided by the Brain Bank Center Wuerzburg, a member of the BrainNet Europe Brain Bank Consortium Network (http://www.brainnet-europe.org/). Human brains were obtained with the consent of the next of kin and according to the guidelines of the National and Local Ethics Committees (Ethical Approval No. 78/99).

## 3. Results

### 3.1. Skin Pigmentation of PD Patients and Controls

In total, 6035 melanin-dominated fluorescence spectra of pigmented skin areas (normal pigmented skin and clinically benign nevi) from PD patients were analyzed, and their spectral profiles were compared to representative spectra from a database of healthy controls. This comparison revealed no obvious differences in the intensity of the melanin-dominated fluorescence between PD patients and controls. A further significant finding is that there were no differences in the spectral shapes of these fluorescence spectra of both normal pigmented skin and nevi (benign and dysplastic) between PD patients and controls. Due to the sensitivity of fluorescence in terms of both the valence electron structure of the fluorophore (melanin) and the composition of the melanin microenvironment in the melanosomes, this coincidence of spectral classes suggests that cutaneous pigmentation has the same composition in Parkinson patients as in controls, both in the normal skin and in melanotic lesions and melanomas. As an example of the measurement, Figure 4a shows a measuring grid for a nevus of a PD patient. The crosses mark the locations of the fluorescence analysis. White crosses mark healthy microareas (spectra of classes 3 or 4 of Figure 2) and the four colored crosses mark three microareas with incipient malignant melanocytic degeneration (class 2) and one single microarea with a fluorescence spectrum typical for melanoma cells (class 1). With a total of only four microareas with spectra of the classes 1 and 2, the nevus shown is regarded as slightly atypical in the clinical sense. Because of the identity of the spectral characteristics of the fluorescence between PD patients and controls, the automatic evaluation given by the device derma FC can be used for the nevus shown. This automatically calculated score is only one-quarter of the limit value for a recommendation for excision. Figure 4b shows representative melanin-dominated spectra from this nevus. The PD patient to whom the nevus of Figure 4a belongs is in dermatological supervision. All remaining 18 studied nevi showed at most three microareas with spectra of classes 2 and 1, meaning cells of a dysplastic nevus and melanoma, respectively. Altogether, a total of 13 microareas with spectra of type 1 and 35 microareas of type 2 were measured. No fluorescence spectra other than classes 1 to 4 were observed (exception: well-characterized fluorescence of hair shafts in a total of 2% of the measured 6035 microareas). In summary, this shows that the cutaneous pigmentation of PD patients had the same melanin properties as seen in controls. Notably, the malignant melanocytic degeneration of nevi toward melanoma was also characterized by the spectra of classes 2 and 1.

### 3.2. Neuromelanin Fluorescence of SNpc from PD and Controls

The overall result of a fluorescence measurement on an FFPE preparation of the postmortem SNpc of PD patients is shown in Figure 5.

Examples of two-photon-excited fluorescence spectra of SNpc (i.e., spectra from the individual measurement areas) of deceased non-PD controls are shown in Figure 6.

When the NAD(P)H spectrum is subtracted from the (green-lined) two-component spectra, the spectrum of the NM in the SNpc of non-PD patients is obtained (Figure 7, blue line). That these different spectra always lead to the same spectral profile (pure NM) confirms that only two components contribute to the total fluorescence. Similarly, two-photon-excited fluorescence spectra of postmortem SNpc of PD patients result in the spectrum of NM in the SNpc of PD (Figure 7, yellow line). This shows that the spectral profiles of NM in the postmortem SNpc of Parkinson patients and controls in the range between 470 nm and 650 nm are identical. This applies to all SNpc samples examined here.

This means that, during the course of PD, the fluorescence in the spectral region attributable to the NM π-electron system remains unchanged. Since fluorescence is generally a sensitive indicator of compositional/structural changes of the fluorophore, it is suggested that NM degradation in PD progress, as shown with this method, occurs without conformational change in the NM π-electron structure. The spectral range below 470 nm (Figure 7) lends itself to further investigations, e.g., for metal incorporation.

To our knowledge, Figure 7 shows for the first time a fluorescence spectrum of NM in FFPE. On the one hand, it is distinctly different from all melanin fluorescence spectra in healthy human skin tissue (normally pigmented nevi) in vivo and in the FFPE preparation. On the other hand, it is largely similar to the melanin fluorescence of melanoma. This is particularly evident when calculating the different spectra analogous to the procedure described in Figure 6 and Figure 7 (not shown here).

## 4. Discussion

The melanin-dominated fluorescence spectra of normal pigmented skin and clinically benign nevi (including those with slight atypia) of PD patients show no difference to the corresponding spectra of PD-free controls of patients with Caucasian skin type (Fitzpatrick 2 and 3, respectively). In these spectra obtained from the cutaneous pigmentation of PD patients, no fingerprint of PD and no sign of increased MM predisposition could be observed. The results presented here indicate rather a pigmentation-independent link between PD and MM for Caucasians, whereby one can neither prove nor exclude this hypothesis.

However, the present study did not include PD patients of red hair color (RHC) phenotype. In relation to other hair colors, the RHC phenotype carries a higher MM risk and a higher pheo-/eumelanin ratio due to loss-of-function *MC1R* variants (including, for instance, p.R 151C), which facilitate pheomelanin formation [28]. Indeed, statistical results of our collected data (not shown) indicate an increased PD risk of healthy RHC individuals, which was also suggested by other authors [4,27]. Recent studies revealed a UV-independent pathway to MM genesis based on pheomelanin [29], which is of particular interest against the background of pheomelanin as an NM component. Echogenic SNpc in investigations of controls with different skin types (Fitzpatrick 1 to 5), which also show an increasing PD risk of lighter-skinned types, point in the same direction of a potential pathogenic role of pheomelanin [30].

## 5. Conclusions

Recent publications pointed out that preventive clinical examinations for MM are particularly desirable in PD patients [1,3,31]. Although the present studies show that there is no specific fluorescence fingerprint of an increased MM risk of Parkinson patients, they can exhibit specific fluorescence spectra of incipient malignant melanocytic degeneration. A dermatofluoroscopic screening of suspicious pigmentary lesions is, therefore, recommended especially for PD patients.

## Figures and Tables

**Figure 1 cells-08-00592-f001:**
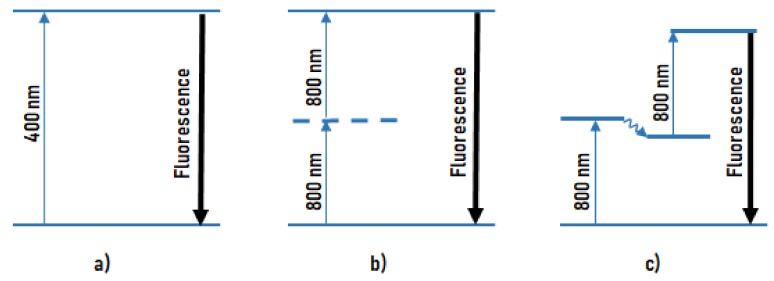
The principle of dermatofluoroscopy. Different modes of excitation of fluorescence in the visible spectral region: (**a**) conventional fluorescence excitation by one photon (e.g., 400 nm); (**b**) excitation by simultaneous absorption of two photons via an only virtual energy level (e.g., 800 nm, preferably from a femtosecond laser); (**c**) excitation by stepwise absorption of two photons (e.g., 800 nm, preferably from a nanosecond laser) via a real intermediate energy level.

**Figure 2 cells-08-00592-f002:**
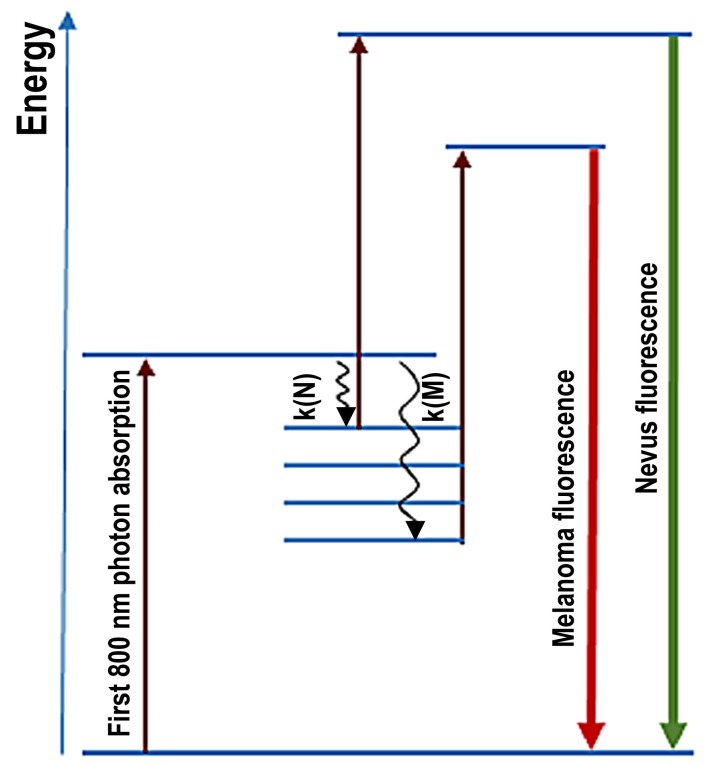
Simplified energy level scheme of melanin in melanosomes of nevi and of melanoma illustrates the process responsible for fluorescence spectra in the excited state: radiationless relaxation in nevi k (N) and in melanoma k (M). The relaxation includes ultrafast Franck–Condon relaxation (tuning of the core configuration to the changed excited-state electron distribution) and nonradiative vibrational relaxation.

**Figure 3 cells-08-00592-f003:**
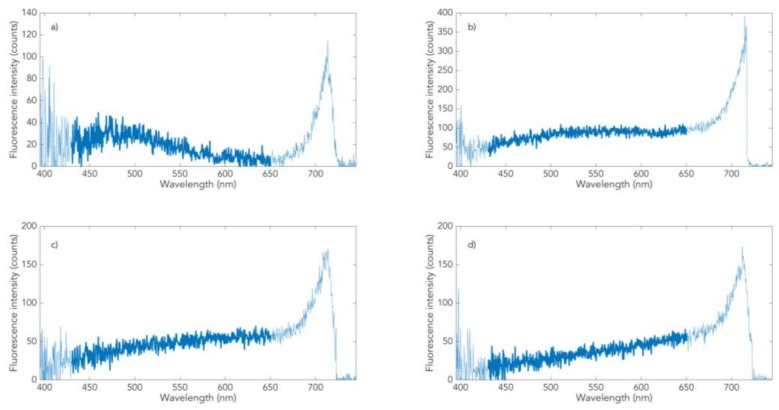
Dermatofluoroscopy of skin from Caucasian patients (Fitzpatrick type 2, 3) in vivo. The four representative classes of melanin-dominated spectra: (**a**) normal pigmented skin (class 4), (**b**) benign nevus (class 3), (**c**) dysplastic nevus (class 2), and (**d**) melanoma (class 1).

**Figure 4 cells-08-00592-f004:**
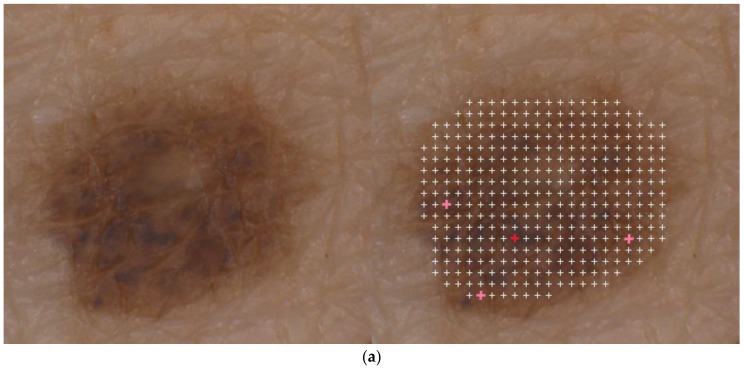
Dermatofluoroscopy of a nevus from a Parkinson patient: (**a**) location of the measuring grid on the nevus, the grid pitch is 200 µm; (**b**) representative melanin-dominated spectra of this nevus. Green line: fluorescence from nevomelanocytes of a benign nevus area. Yellow line: fluorescence corresponding to that from nevomelanocytes of a dysplastic nevus area. Red line: fluorescence corresponding to that from melanoma cells. The wavelength (horizontal line) is given in nanometers (nm), counts of fluorescence intensity on the vertical line.

**Figure 5 cells-08-00592-f005:**
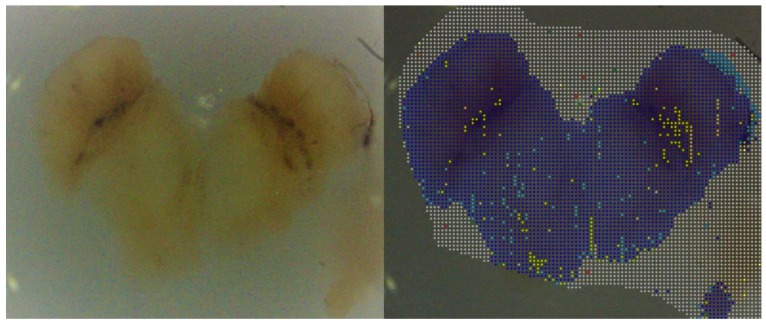
Dermatofluoroscopy of the postmortem substantia nigra pars compacta (SNpc) of a Parkinson patient. The grid above the specimen shows the individual measuring areas; the colored dots indicate the local neuromelanin concentration resulting from the measurements, here for the sake of clarity only as a rough differentiation (blue, no neuromelanin; yellow-green, neuromelanin). Scale 1:5.

**Figure 6 cells-08-00592-f006:**
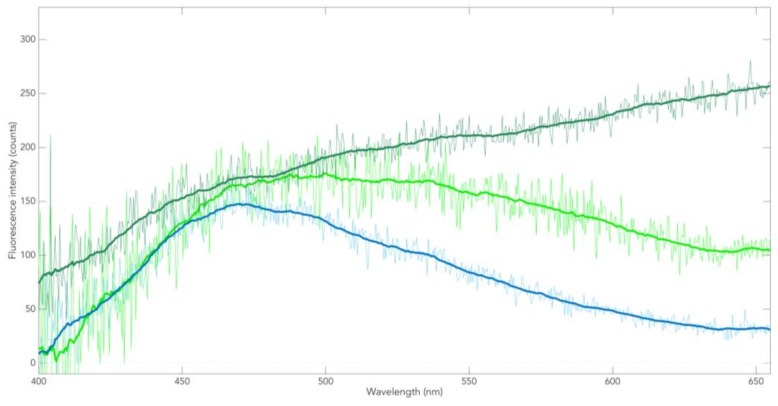
Examples of fluorescence spectra of postmortem SNpc of control case. The blue line represents a spectrum from a largely neuromelanin (NM)-free region; this fluorescence is the tissue autofluorescence from NAD(P)H. The other two spectra represent—from light to dark green—increasing contribution of NM fluorescence to the autofluorescence; the dark-green lined spectrum results from regions with maximum NM content.

**Figure 7 cells-08-00592-f007:**
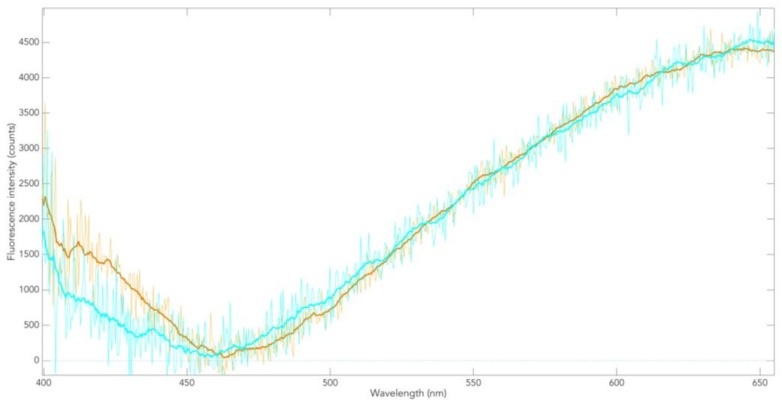
Fluorescence spectrum of NM in the postmortem SNpc of controls (blue line) and in a Parkinson patient (green line).

**Table 1 cells-08-00592-t001:** Overview of the data of the patient cohort and the nevi; F = female, M = male, cl.1 = class 1, cl.2 = class 2.

Patient	Age	Gender	Size of Nevi (mm)	Location of Nevi	Number of Spectra/cl.1-cl.2-Fault
1	62	F	3.0 × 2.0	Sole	160/0-0-3
			2.4 × 3.8	Right lower leg	240/0-0-0
2	39	F	4.0 × 3.8	Abdomen	364/2-1-4
3	72	F	3.2 × 3.8	Right temporal	291/1-2-3
4	78	M	3.8 × 4.0	Chest wall	345/0-2-9
5	50	M	4.6 × 4.0	Right shoulder	350/0-3-7
			3.6 × 4.2	Right upper leg	381/0-3-10
6	52	M	3.8 × 3.6	Right foot	310/2-1-8
			5.4 × 2.6	Back	464/0-3-9
7	67	M	4.4 × 3.6	Abdomen	392/1-3-8
8	75	M	4.5 × 3.0	Left shoulder	310/0-3-10
9	75	M	2.2 × 3.0	Abdomen	158/1-2-2
10	78	M	4.4 × 4.4	Left shoulder	494/2-1-14
11	67	M	4.6 × 3.6	Back	453/0-3-10
12	61	M	3.5 × 3.5	Chest wall	278/0-3-6
13	35	M	2.8 × 3.8	Right temporal	275/2-1-1
			4.6 × 4.8	Back	442/2-1-9
14	53	M	3.0 × 4.0	Chest wall	250/0-0-0
			1.6 × 1.8	Left upper leg	77/0-3-8

**Table 2 cells-08-00592-t002:** Overview of the postmortem brain tissue samples from the midbrain from 10 neuropathologically characterized Parkinson’s disease (PD) cases versus 10 age-matched controls; 1 = controls, 2 = PD cases, F = female, M = male, PMI = postmortem interval in hours, DD = Parkinson’s disease duration in years.

Group	Age	Gender	PMI (h)	DD (years)
1	77	F	9.5	--
1	83	M	15	--
1	76	M	9	--
1	85	M	12.5	--
1	71	F	16	--
1	84	F	19	--
1	79	M	10	--
1	83	M	12.5	--
1	88	F	13	--
1	75	F	17	--
2	83	M	9.5	15
2	77	M	13.5	9
2	85	M	17	12
2	72	M	16.5	6
2	83	F	12	13
2	90	M	19.5	22
2	74	M	15.5	9
2	76	F	12	10
2	88	F	9.5	21
2	81	M	13.5	17

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
