# Peer review of "Melanin and Neuromelanin Fluorescence Studies Focusing on Parkinson’s Disease and Its Inherent Risk for Melanoma"

_cells, 2019, doi:10.3390/cells8060592_

Round 1

Reviewer 1 Report

This original study may shed some light to the intriguing association between Parkinson’s disease (PD) and melanoma (and vice versa). While the presented results could be considered mainly as negative (i.e. no differences are found in the fluorescence spectroscopic pattern between PD and controls) the approach is interesting and worth reporting.

1.       The results are presented only in a qualitative manner. Is it possible to quantify the dermatofluorescence of the different experimental groups and be compared by statistical methods?

2.       Is the fluorescence spectrum of brain neuromelanin similar to skin melanin? This could be an interesting point to be further discussed/developed since it is widely assumed that peripheral melanin and brain neuromelanin are synthesized by different pathways (i.e. enzymatically by tyrosinase in the former; spontaneous oxidation in the latter, although tyrosinase appears also to be expressed at low levels in the brain). In this context, could the similarity of the fluorescence spectra between both pigments indicate a similar synthetic pathway (i.e. enzymatic)?

3.       It may be useful to indicate in a Table the main pathologic features of the pigmented lesions and the features of the PD and control brains analyzed (age, gender, PMI, disease duration, dermatological lesions, etc.).

Author Response

Reviewer 1

We are grateful for the suggestions and criticism of the reviewers and we will try to answer to all the issues raised (we marked the changes and additions in the manuscript in red).

.

This original study may shed some light to the intriguing association between Parkinson’s disease (PD) and melanoma (and vice versa). While the presented results could be considered mainly as negative (i.e. no differences are found in the fluorescence spectroscopic pattern between PD and controls) the approach is interesting and worth reporting.

1. The results are presented only in a qualitative manner. Is it possible to quantify the dermatofluorescence of the different experimental groups and be compared by statistical methods?

Answer: We added the following phrase (line 210) regarding the algorithm of this analysis and a new reference (No. 23 in the new references list):

RMSD has a fixed upper limit, and spectra which exceed this limit elude this classification (e.g. hairs, marker fluorophores or impurities). Further details of the automated assignment of the spectra measured with derma FC are previous described (Forschner et al, 2018).

2.  Is the fluorescence spectrum of brain neuromelanin similar to skin melanin? This could be an interesting point to be further discussed/developed since it is widely assumed that peripheral melanin and brain neuromelanin are synthesized by different pathways (i.e. enzymatically by tyrosinase in the former; spontaneous oxidation in the latter, although tyrosinase appears also to be expressed at low levels in the brain). In this context, could the similarity of the fluorescence spectra between both pigments indicate a similar synthetic pathway (i.e. enzymatic)?

Answer: We supplemented the following sentence and text:

Line 337: This shows that the spectral profiles of NM in the postmortem SNpc of Parkinson patients and controls in the range between 470 nm and 650 nm are identical.

Line 342: This means that during the course of PD the fluorescence in the spectral region attributable to the NM π-electron system remains unchanged.
Since fluorescence is generally a sensitive indicator of compositional/structural changes of the fluorophore, it is suggested that NM degradation in PD progress, as shown with this method, occurs without conformational change in the NM π electron structure. The spectral range below 470 nm (Fig. 7) lends itself to further investigations, e.g. for metal incorporation.
To our knowledge, Fig. 7 shows for the first time a fluorescence spectrum of NM in FFPE. On the one hand it is distinctly different from all melanin fluorescence spectra in healthy human skin tissue (normally pigmented naevi) in vivo and in the FFPE preparation. On the other hand, it is largely similar to the melanin fluorescence of melanoma. This is particularly evident when calculating the different spectra analogous to the procedure described in Fig. 6 and 7 (not shown here).

3. It may be useful to indicate in a Table the main pathologic features of the pigmented lesions and the features of the PD and control brains analyzed (age, gender, PMI, disease duration, dermatological lesions, etc.).

Answer: We supplemented the following tables in line 241, respectively line 262:

Patient

Age

Gender

Size of naevi (mm)

Location of naevi

Number   of spectra/cl.1-cl.2-fault

1

62

f

3,0 x 2,0

sole

160/0-0-3

2,4 x 3,8

right lower leg

240/0-0-0

2

39

f

4,0 x 3,8

abdomen

364/2-1-4

3

72

f

3,2 x 3,8

right temporal

291/1-2-3

4

78

m

3,8 x 4,0

chest wall

345/0-2-9

5

50

m

4,6 x 4,0

right shoulder

350/0-3-7

3,6 x 4,2

right upper leg

381/0-3-10

6

52

m

3,8 x 3,6

right foot

310/2-1-8

5,4 x 2,6

back

464/0-3-9

7

67

m

4,4 x 3,6

abdomen

392/1-3-8

8

75

m

4,5 x 3,0

left shoulder

310/0-3-10

9

75

m

2,2 x 3,0

abdomen

158/1-2-2

10

78

m

4,4 x 4,4

left shoulder

494/2-1-14

11

67

m

4,6 x 3,6

back

453/0-3-10

12

61

m

3,5 x 3,5

chest wall

278/0-3-6

13

35

m

2,8 x 3,8

right temporal

275/2-1-1

4,6 x 4,8

back

442/2-1-9

14

53

m

3,0 x 4,0

chest wall

250/0-0-0

1,6 x 1,8

left upper leg

77/0-3-8

Table 1: Overview of the data of the patient cohort and the naevi; f = female, m = male, cl.1 = class 1, cl.2 = class 2

Group

Age

Gender

PMI (h)

DD

1

77

f

9,5

--

1

83

m

15

--

1

76

m

9

--

1

85

m

12,5

--

1

71

f

16

--

1

84

f

19

--

1

79

m

10

--

1

83

m

12,5

--

1

88

f

13

--

1

75

f

17

--

2

83

m

9,5

15

2

77

m

13,5

9

2

85

m

17

12

2

72

m

16,5

6

2

83

f

12

13

2

90

m

19,5

22

2

74

m

15,5

9

2

76

f

12

10

2

88

f

9,5

21

2

81

m

13,5

17

Table 2: Overview of the postmortem brain tissue samples from the midbrain from 10 neuropathologically characterized PD cases versus 10 aged maged controls 1 = controls, 2 = PD cases, f = female, m = male, PMI = postmortem interval in hours, DD = Parkinson disease duration in years

Furthermore we added and modified as following:

We redrafted the legend of Figure 4:

Line 306: Figure 4: Dermatofluoroscopy of a naevus from a Parkinson patient: a) location of the measuring grid on the naevus, the grid pitch is 200 µm; b) representative melanin-dominated spectra of this naevus. Green line: fluorescence from nevomelanocytes of a benign naevus area. Yellow line: fluorescence corresponding to that from naevomelanocytes of a dysplastic naevus area. Red line: fluorescence corresponding to that from melanoma cells.

 Line 392: This publication was funded by the German Research Foundation (DFG) and the University of Wuerzburg in the funding programme Open Access Publishing.

Reviewer 2 Report

My biggest concern with this paper is in what I feel is a bit too ambitious of a conclusion that appears on lines 283-284 "The results presented here clearly indicate a pigmentation-independent link between PD and MM for Caucasians." I'm fine with saying that the current study was unable to detect such a link, but not so sure they have established that such a link does not exist.

The spectral classes presented are not drastically different. It is not clear to me whether the shape of the curve represents something real or simply altered concentrations of various fluorophores. In control skin, there appears to be a larger relative ~450 nm contribution (fig 3 panel A), but it is not obvious whether this is because the melanin is lower in concentration such that NADH fluorescence predominates, while the flattened curve of melanoma (panel D) simply has relatively more melanin. From the methods provided and discussion, it seems there is no possibility of quantification??

One conclusion is that PD patients and controls show no difference in pigmentation during disease progression. To my eye however, the curves in figure 7 are not identical. Can the authors help me understand the peak near 435 nm and other artifacts? Some serious discussion on the limitations of the technique I believe is warranted here. Is it actually sensitive enough to detect changes in neuromelanin structure? Oxidation state? Metal incorporation? Bound dopamine or other amines? Free or vesicle enclosed? There are simply too many variables regarding possible contributors to the fluor signal for me to be comfortable with this conclusion.

Figure 5 needs some better discussion I believe. In particular, the rather obvious areas of dark melanin pigment do not correspond with the areas containing neuromelanin as detected by the two-photon excitation. There are areas that appear light yet the data show strong melanin fluor signal from them (bottom left lobe of fig 5, right panel). It is concerning that the detection method does not appear to correlate with visible pigmentation, and this needs to be addressed. Are the dark areas not all neuromelanin? If not, what are they? If light areas are showing neuromelanin, does this imply the method can detect diffuse, un-granulated polymer? Help me understand this better.

Line 95 states that no previous studies have noted fluorescence from neuromelanin. This should probably be clarified to note that previous in vitro studies have in fact found fluorescence (notably, upon nicotine binding), but no in situ work has be published previously.

Author Response

Reviewer 2

We are grateful for the suggestions and criticism of the reviewers and we will try to answer to all the issues raised (we marked the changes and additions in the manuscript in red).

My biggest concern with this paper is in what I feel is a bit too ambitious of a conclusion that appears on lines 283-284 "The results presented here clearly indicate a pigmentation-independent link between PD and MM for Caucasians." I'm fine with saying that the current study was unable to detect such a link, but not so sure they have established that such a link does not exist.

Answer: We supplemented in line 361 the following sentence:
The results presented here indicate rather a pigmentation-independent link between PD and MM for Caucasians, whereby one can neither prove nor exclude this hypothesis.

The spectral classes presented are not drastically different. It is not clear to me whether the shape of the curve represents something real or simply altered concentrations of various fluorophores. In control skin, there appears to be a larger relative ~450 nm contribution (fig 3 panel A), but it is not obvious whether this is because the melanin is lower in concentration such that NADH fluorescence predominates, while the flattened curve of melanoma (panel D) simply has relatively more melanin. From the methods provided and discussion, it seems there is no possibility of quantification??

Answer: We added the following text and sentences:

Line 169: The spectrum of normally pigmented skin (Fig. 3 a) is a superposition of melanin fluorescence with a maximum at about 500 nm with the NAD(P)H-dominated autofluorescence (maximum at about 470 nm, obtained from oculo-cutaneous skin tissue). The ratio of the two components determines the position of the resulting fluorescence maximum [20]; on low pigmented Fitzpatrick skin type 1 both bands appear separately (D. Leupold et al, submitted). Melanin fluorescence from the naevomelanocytes of benign or dysplastic naevi (Fig. 3b, c) shows a distinctly different, flatter and red-shifted spectral profile than fluorescence from melanocytes. An obvious cause of the difference could be the lacking discharge of melanosoma out of melanocytes (shutdown of the dominant influence of keratinocytes) and the changing structural alignments of the melanin p-systems that determine fluorescence („p-stacking“). The red shift of the fluorescence spectrum changes into the characteristic curve according to Fig. 3d in melanoma: a constant increase in intensity from 440 nm to 650 nm. (Mathematically described, this means that the first derivative of the spectral fluorescence course dIF (lambda) / d (lambda) for melanoma is a horizontal straight line between 440 and 650 nm. The spectral fluorescence course of dysplastic nevi (Fig.3c) differs from that of melanoma by a reduced increase above about 570 nm (this means a drop in the constant value of the first derivative above 570 nm). Benign naevi are characterized by a zero crossing of the first derivative in the range between about 530 and 550 nm).

Line 210: RMSD has a fixed upper limit, and spectra which exceed this limit elude this classification (e.g. hairs, marker fluorophores or impurities). Further details of the automated assignment of the spectra measured with derma FC are described (Forschner et al, 2018)..

Line 296: No fluorescence spectra other than class 1 to 4 were observed (exception: well-characterized fluorescence of hair shafts in a total of 2% of the measured 6035 microareas).

One conclusion is that PD patients and controls show no difference in pigmentation during disease progression. To my eye however, the curves in figure 7 are not identical. Can the authors help me understand the peak near 435 nm and other artifacts? Some serious discussion on the limitations of the technique I believe is warranted here. Is it actually sensitive enough to detect changes in neuromelanin structure? Oxidation state? Metal incorporation? Bound dopamine or other amines? Free or vesicle enclosed? There are simply too many variables regarding possible contributors to the fluor signal for me to be comfortable with this conclusion.

Answer: We supplemented the following sentence and text:

Line 337: This shows that the spectral profiles of NM in the postmortem SNpc of Parkinson patients and controls in the range between 470 nm and 650 nm are identical.

Line 342:
This means that during the course of PD the fluorescence in the spectral region attributable to the NM π-electron system remains unchanged. Since fluorescence is generally a sensitive indicator of compositional/structural changes of the fluorophore, it is suggested that NM degradation in PD progress, as shown with this method, occurs without conformational change in the NM π electron structure. The spectral range below 470 nm (Fig. 7) lends itself to further investigations, e.g. for metal incorporation.
To our knowledge, Fig. 7 shows for the first time a fluorescence spectrum of NM in FFPE. On the one hand it is distinctly different from all melanin fluorescence spectra in healthy human skin tissue (normally pigmented naevi) in vivo and in the FFPE preparation. On the other hand, it is largely similar to the melanin fluorescence of melanoma. This is particularly evident when calculating the different spectra analogous to the procedure described in Fig. 6 and 7 (not shown here).

Figure 5 needs some better discussion I believe. In particular, the rather obvious areas of dark melanin pigment do not correspond with the areas containing neuromelanin as detected by the two-photon excitation. There are areas that appear light yet the data show strong melanin fluor signal from them (bottom left lobe of fig 5, right panel). It is concerning that the detection method does not appear to correlate with visible pigmentation, and this needs to be addressed. Are the dark areas not all neuromelanin? If not, what are they? If light areas are showing neuromelanin, does this imply the method can detect diffuse, un-granulated polymer? Help me understand this better.

Answer: The yellow-green dots on the right panel indicate indeed neuromelanin as detected by the two-photon excitation. On the bottom left lobe (and also near of median line) there are some dots indicating neuromelanin. This can be explained by the presence of individual neuromelanin-containing neurons aside the SN (Mai, J., Paxinos, G., The human nervous system: Elsevier Academic Press. 2011, ISBN: 9780123742360). Not necessarily all cells are neurons, but also microglial cells: “NM is sometimes found within glial cells, but there is no evidence that this pigment is produced within these cells. As NM-containing glia occurs more frequently in the aged brain, it is thought that the pigment is phagocytosed into glial cells for removal from the brain following its release from degenerating NM-pigmented neurons within the substantia nigra (Double et al, The comparative biology of neuromelanin and lipofuscin in the human brain, Cell Mol Life Sci, 2008)”.

Line 95 states that no previous studies have noted fluorescence from neuromelanin. This should probably be clarified to note that previous in vitro studies have in fact found fluorescence (notably, upon nicotine binding), but no in situ work has be published previously.

Answer: We added in line 105 the term “in situ” and the following sentence:
To our knowledge, no fluorescence spectra excited by UV or visible radiation have previously been found for NM in situ. However, a weak fluorescence of synthetic NM is reported, enhanced by nicotine binding (Haining et al, 2016).

Furthermore we added and modified as following:

We redrafted the legend of Figure 4:

Line 306: Figure 4: Dermatofluoroscopy of a naevus from a Parkinson patient: a) location of the measuring grid on the naevus, the grid pitch is 200 µm; b) representative melanin-dominated spectra of this naevus. Green line: fluorescence from nevomelanocytes of a benign naevus area. Yellow line: fluorescence corresponding to that from naevomelanocytes of a dysplastic naevus area. Red line: fluorescence corresponding to that from melanoma cells.

 Line 392: This publication was funded by the German Research Foundation (DFG) and the University of Wuerzburg in the funding programme Open Access Publishing.

Round 2

Reviewer 1 Report

The Authors made an effort to answer the reviewer's comments satisfactorily.

Reviewer 2 Report

I appreciate your addressing my concerns and have none further to add.